# Auto-Calibrated Charge-Sensitive Infrared Phototransistor at 9.3 µm

**DOI:** 10.3390/s23073635

**Published:** 2023-03-31

**Authors:** Mohsen Bahrehmand, Djamal Gacemi, Angela Vasanelli, Lianhe Li, Alexander Giles Davies, Edmund Linfield, Carlo Sirtori, Yanko Todorov

**Affiliations:** 1Laboratoire de Physique de l’Ecole Normale Supérieure, Ecole Normale Supérieure, Paris Sciences et Lettres, Centre National de la Recherche Scientifique (CNRS), Université de Paris, 24 Rue Lhomond, 75005 Paris, France; 2School of Electronic and Electrical Engineering, University of Leeds, Leeds LS2 9JT, UK

**Keywords:** mid-infrared spectral domain, quantum detector, charge-sensitive infrared phototransistor (CSIP), detector performance

## Abstract

Charge-sensitive infrared photo-transistors (CSIP) are quantum detectors of mid-infrared radiation (λ=4 µm−14 µm) which have been reported to have outstanding figures of merit and sensitivities that allow single photon detection. The typical absorbing region of a CSIP consists of an Al_x_Ga_1-x_As quantum heterostructure, where a GaAs quantum well, where the absorption takes place, is followed by a triangular barrier with a graded x(Al) composition that connects the quantum well to a source-drain channel. Here, we report a CSIP designed to work for a 9.3 µm wavelength where the Al composition is kept constant and the triangular barrier is replaced by tunnel-coupled quantum wells. This design is thus conceptually closer to quantum cascade detectors (QCDs) which are an established technology for detection in the mid-infrared range. While previously reported structures use metal gratings in order to couple infrared radiation in the absorbing quantum well, here, we employ a 45° wedge facet coupling geometry that allows a simplified and reliable estimation of the incident photon flux *Φ* in the device. Remarkably, these detectors have an “auto-calibrated” nature, which enables the precise assessment of the photon flux *Φ* solely by measuring the electrical characteristics and from knowledge of the device geometry. We identify an operation regime where CSIP detectors can be directly compared to other unipolar quantum detectors such as quantum well infrared photodetectors (QWIPs) and QCDs and we estimate the corresponding detector figure of merit under cryogenic conditions. The maximum responsivity R = 720 A/W and a photoconductive gain G~2.7 × 10^4^ were measured, and were an order of magnitude larger than those for QCDs and quantum well infrared photodetectors (QWIPs). We also comment on the benefit of nano-antenna concepts to increase the efficiency of CSIP in the photon-counting regime.

## 1. Introduction

Currently, the most widespread quantum detectors of radiation in the mid-infrared region (λ=4 µm−14 µm) are either unipolar detectors based on III–V quantum well heterostructures, such as QWIP (quantum well infrared photodetector) [1] and QCD (quantum cascade detectors) [2,3] or photodiodes based on small bandgap materials, such as HgCdTe (mercury cadmium telluride) [4,5]. The QCDs are particularly appealing, as they rely on the quantum engineering of the electronic transport in tunnel-coupled quantum wells, an approach also used for radiation sources such as QCLs (quantum cascade lasers) [6]. For photoconductive detectors, such as QWIPs, the performance is expressed through the detector responsivity R[A/W]=(e/ћω)×η×G, where *e* is the electron charge, ћω is the energy of the photons to be detected, *η* is the overall absorption efficiency of the detector and *G* is the photoconductive gain; that is, the number of electrons circulating in the detector read-out circuit per one absorbed photon [7]. By definition, *η*
≤ 1, and although *G* can exceed unity for QWIPs [8], typically for QCDs, *G* is normally much smaller than 1, as it scales as *G* = 1/*N*, where *N* is the number of detector periods. Assuming *ηG*
≤ 1, a theoretical upper bound for the responsivity of such detectors is set by e/ћω which is typically 10 A/W for a photon wavelength λ=9 µm. In 2004, the Komiyama group from the University of Tokyo proposed a detector concept where *G* can have values much larger than one [9,10,11]. These detectors, called charge-sensitive infrared phototransistors, CSIPs, are an analogue to the single electron transistors from mesoscopic physics [12]. Moreover, they use the same quantum-engineering as QCDs and QCLs. In these devices, the photon absorption promotes an electron transfer from a lateral gate-insulated quantum well, forming an island, into a source-drain (SD) channel. The positive charges that remain in the island induce a constant electric field that changes the source-drain conductance in a similar way to a field-effect transistor [13]. The photocurrent signal is the extra source-drain current $\Delta I_{SD}$ provided by the photoexcited electrons. The positive vacancies (holes) of the Fermi gas have a very long lifetime, $\tau_{h}$, which allows the CSIP device to operate in a transient regime, where one can monitor the time derivative of the source-drain current ${dI}_{SD}/dt$. This quantity, at early time t, is directly proportional to the total number of photon *Φ* incidents on the sample per unit time [11]. In this regime, CSIPs have been demonstrated to act as photon counters, if the rate of impinging photons is sufficiently small [14,15]. In the transient regime, the detector gain on the order of *G*~10^7^ and responsivities in the order of R~10^3^–10^6^ A/W have been reported [10]. A more direct comparison with QWIPs and QCDs can be obtained in the so-called “saturation regime” (see Ref. [11]), where a dynamic equilibrium is reached between the transferred electrons and the recombination of positive vacancies in the lateral gate-insulated island. The extra source drain current $\Delta I_{SD}$ is then proportional to the incident photon flux *Φ* and the detector gain can be expressed as $G=\tau_{h}/\tau_{transit}$, where $\tau_{transit}$ is the transit time for an electron travelling between the source and drain. As the recombination rate of holes ($1/\tau_{h}$) can be very short, we typically have $\tau_{h}{\gg \tau }_{transit}$ and, therefore, the detector responsivity can reach values as high as 720 A/W in the regime of weak photon fluxes (see below for an example), which is still an impressive value, compared to other quantum detectors without any amplifier circuits.

CSIP detectors have been employed in infrared confocal [16] and near-field microscopes, for wavelengths λ = 14.7 µm and λ = 10.2 µm [17,18]. In this wavelength range, the near-field imaging systems are sensitive to the thermal radiation emitted by nanoscale objects. Such system have proven to be a powerful tool for the investigation of heat transfer phenomena at the nanoscale, such as hot electron thermalization in constricted two dimensional electron gas [19,20], Joule heating in nanostructured conductors [21] and thermally excited evanescent fields in an electrically biased graphene [22]. Furthermore, the CSIP device architecture itself offers many degrees of freedom, such as the possibility to implement dynamically controlled optical gates in order to reduce the effect of ambient thermal radiation [23], as well as the implementation of CSIPs in large arrays [24], and engineering optimized light collection [25]. Dual color CSIPs at λ =10.6 µm and 15.7 µm have also been recently demonstrated thanks to a judicious design of the quantum well heterostructure [26].

Here, we report a CSIP device designed to detect infrared radiation with a wavelength of 9.3 µm. The CSIP was realized with a gallium arsenide/aluminum-gallium arsenate GaAs/Al_0.33_Ga_0.67_As material system. The majority of the CSIP devices reported in the literature employ a graded Al_x_Ga_1-x_As barrier that connects the absorbing quantum well and the SD channel where x is varied linearly in a typical range x = 0.13–0.08 (see, for instance, Ref. [26]). Here, we employed a heterostructure with a constant x = 0.33, making the absorbing region design conceptually closer to that of QCD and QWIP detectors [1,2,3]. Furthermore, we explored a simple light-coupling scheme, based on a 45° wedged substrate, that is also widely used with unipolar detectors [27]. This configuration allows a simple and reliable estimation of the quantum efficiency of the absorbing region, and allows the analysis of the performances of the CSIPs with respect to other quantum detectors. We also provide an expression of the incident photon flux measured by the detector in terms of the size of the SD channel and absorption cross-section of the intersubband transition previously introduced in Ref. [28] (Equation (4)). We underline thus the “auto-calibrated” nature of such detectors, which enable precise photon counting simply by measuring the electrical characteristics and from knowledge of the device geometry. These results show the quantitative way in which photonic architectures with antenna functions can be used to optimize the CSIP performance.

## 2. Electrical and Optical Characteristics of the Detector

In Figure 1a, we describe the quantum design of the CSIP structure which is based on a sequence of GaAs/Al_0.33_Ga_0.67_As wells/barriers. It consists of a 7.2 nm upper quantum well (UQW) where the optical transition takes place between subbands 1 and 2. As shown in the figure, subband 2 is strongly delocalized in the section which consists of a sequence of thin barriers and wells. The sequence has been designed to simulate a triangular potential ramp as required for CSIP [9,10,11]. After optical excitation, electrons relax through the subbands of the triangular region down to the level 1′ of the rightmost 36 nm wide quantum well (LQW) which plays the role of the transistor source-drain channel. A full band structure simulation, including the Schottky barrier on the left, is shown in Figure 1b. Electrons are provided through two 50 nm wide Si-doped regions at a density of 10^18^ cm^−3^ on both sides of the heterostructure. Our estimation of the number of active charges in the UQW and LQW is discussed further. The optical transition of the structure was probed in multi-pass transmission experiments performed at 4K, as shown in the inset to Figure 1c [27]. For this experiment, the structure is illuminated through the substrate that was wedged at 45°. A large TiAu gate and two lateral annealed ohmic contacts are processed on the top of the structure, which allowed a differential absorption experiment, where only the charge density of the UQW is varied. For this experiment, the infrared radiation is provided by a Globar lamp of a Fourier transform infrared interferometer (FTIR). The Globar acts as a blackbody source with a typical temperature of 1200 °C. The infrared beam is collimated at the exit of the FTIR and is focused on one of the wedged facets of the sample with the help of a metallic F1 parabolic mirror. The signal at the existing facet is collected from another F1 parabolic mirror and refocused on the commercial HgCdTe detector connected to a lock-in amplifier. A 20 kHz square wave voltage with V_min_ = −10 V and V_max_ = 0 V with 50% duty cycle is applied between the TiAu gate and the ohmic contacts, which results in a change in the UQW charge density: at large negative bias, the UQW is almost depleted. The signal from the lock-in amplifier, which is referenced on the square wave voltage, thus corresponds to the difference between the sample transmission for fully charged UQW and for a depleted state. The resulting differential signal from the lock-in is fed back to the FTIR in order to obtain an interferogram in a step-scan mode. The resulting spectra, shown in Figure 1c correspond to the absorption of the 1→2 intersubband transition, which has a peak value at the expected energy ΔE=133 meV (λ = 9.3 µm), with a FWHM ℏΓ= 9.4 meV, as extracted from a Lorentzian fit of the data (dotted curve on Figure 1c).

As shown in Figure 2a,c, the detector has been processed in a transistor-like architecture, following Ref. [11]. Figure 2a is an optical photograph of a typical device, which has been defined by a chemical etch. It consists of an SD channel of width *W* and length *L* that are varied between devices; for the device of Figure 2a, W=50 µm and L=100 µm. The S and D contacts are obtained by annealed AuGeNi contacts. The device is processed with two gates. The purpose of the floating gate, FG, is to ensure the coupling of the incident infrared radiation into the UQW. In the structures reported so far, the FG is shaped into a one-dimensional [26] or two dimensional diffraction grating [23] in order to allow normal incidence coupling. Such structures require careful numerical modelling in order to estimate the fraction of incident light absorbed by the electronic transition. In the present case, we used a coupling configuration through the 45° wedge substrate, similar to the one illustrated in Figure 1c. We then used a continuous metallic pattern for the FG which covers the absorbing area; in that case, its role is to ensure the correct boundary condition for the electric field of the incident wave that maximizes optical absorption in the UQW [27]. The absorption quantum efficiency then has a simple analytical form, as seen later (Equation (1)).

As seen in Figure 2b, the SD contacts connect both the UQW and LQW. The insulation gate, IG, is used to trap the carriers in the UQW in three dimensions [11]. The RG allows the SD channel to be connected to additional diffused contacts, RC (reset contacts), which makes it possible to restore charge in the UQW [11].

We performed an electrical characterization of the device in cryogenic conditions, at temperature *T* = 10 K. The devices were mounted on the cold finger of a commercial cryostat which was customized with a temperature shield in order to decouple the ambient 300 K blackbody radiation. In Figure 2c, we show the SD current $I_{SD}$ as a function of $V_{IG}$ and for a fixed bias $V_{SD}=50\;mV$. Measurements were taken at 4 K. As expected, we observe two regimes of conductivity: in the region $-0.5V<V_{IG}<0.0V$ both UQW and LQW participated in the conduction. For $V_{IG}<-0.5V,$ the UQW is insulated, and we see a very clear change in the transconductance ${dI}_{SD}/dV_{IG}$. Finally, for sufficiently negative biases, $V_{IG}<-1.4V$, both the UQW and LQW are completely blocked. There is an important change with respect to the RG bias that is due to the photon absorption, which will be commented upon further. As shown in Figure 2d, we have used the electrical characteristics in Figure 2c together with band-structure modelling (Figure 1b) to estimate the charge densities $n_{UQW}$ and $n_{LQW}$ in our structures. For this purpose, we start with initial values $n_{UQW}$ and $n_{LQW}$ at $V_{IG}=0$ V and vary the $V_{IG}$ applied on the left metal contact in Figure 1b. We have determined the pair $n_{UQW}=2.0\times 10^{11}{cm}^{-2}$ and $n_{LQW}=2.5\times 10^{11}{cm}^{-2}$ for which the UQW and the LQW become depleted for $V_{IG}=-0.5V$ and $V_{IG}=-1.4V$, respectively. Hence, together with the absorption measurements from Figure 1c, we determine the absorption efficiency of the UQW using Equation (21) from Ref. [27]:(1)$$\eta =0.15\;n_{UQW}{[10}^{12}{cm}^{-2}]\frac{2f}{\Gamma /2[meV]}\frac{{sin}^{2}(45{}^{\circ})}{cos(45{}^{\circ})}=0.8\%$$

Here, we used an additional factor of 2 to take into account the proximity of the metal layers that enhance the absorption of the UQW [27]. The last factor in the right-hand side arises from the intersubband selection rule and the 45° facet geometry [27]. The oscillator strength *f* = 0.9 has been obtained by summing the oscillator strengths between subband 1 and the subbands with transition energies within 10% of the absorption peak.

Further, the expression of the SD current is:(2)$$I_{SD}=e{(\mu }_{UQW}n_{UQW}+\mu_{LQW}n_{LQW})\frac{W}{L}V_{SD}$$

Thus, from the *IV* measurement of Figure 2c we estimate the mobility of the upper and lower quantum wells at 4K, ${µ}_{UQW}=6.5\times 10^{4}{cm}^{2}/V\cdot s$ and ${µ}_{LQW}=3.5\times 10^{4}{cm}^{2}/V.s$ respectively. For this we assume that $V_{IG}=0V$ and the conductivity arises from both UQW and LQW, whereas at $V_{IG}<-0.5V$ it arises only from the LQW. We should note that we observed small day to day variations of the electrical characteristics, probably due to formation of electric field domains and variability in dopant activation [15].

## 3. Detector Figures of Merit

Here, we briefly recall the operation principle of CSIP; for an extensive description one may refer to Ref. [11]. The working mechanism of CSIP is based on the capacitive coupling between the UQW and LQW. As shown in the results from Figure 2, by applying a sufficiently negative IG bias the SD conductance from the UQW is blocked; the IG thus defines an area where the 2D electron gas in the UQW is isolated from the rest of the structure and forms an island. When a photon is absorbed in the UQW, an electron is promoted to the subband 2 and then relaxes to the SD channel (subband 1′, Figure 1a), as a result the SD conductance changes. Assuming that the structure is illuminated by a photon flux *Φ* at t=0, the time evolution of the extra-SD current ${\Delta I}_{SD}$ owing to photon absorption is provided by [11]:(3)$$\frac{d\Delta I_{SD}}{dt}=\eta Tg\Phi I_{p}-\frac{{\Delta I}_{SD}}{\tau }$$

Here $T=4n/(1+n{)}^{2}=0.7$ is the transmission coefficient of the wedge, $g=sin^{2}(45{}^{\circ})/cos(45{}^{\circ})=0.71$ is a geometrical factor that takes into account the intersubband selection rule in the present geometry, $I_{p}=e{µ}_{LQW}V_{SD}/L^{2}$ is the SD current contribution for a single electron, and *τ* is a relaxation time for the positive charges in the island left by photoexcited electrons. This time can be estimated by modelling the backfilling of electrons from the LQW to the UQW [11]. The characteristic time *τ* is not a fixed parameter, but has a strong dependence of the number of positive charges in the UQW, which is proportional to the signal ${\Delta I}_{SD}$ [11]. Indeed, after each electron is transferred, a positive charge is left in the UQW island that generates an electrostatic bias, which lowers the UQW and flattens the effective triangular barrier that separates the UQW from LQW and thus favors the hole recombination. Eventually, for a sufficiently high number of positive charges, the potential barrier will lower to a point where the time τ becomes short enough so that the backfilling of the UQW compensates the optical absorption for a maximal number of positive charges; the device will reach a saturation point. The positive charges accumulated in the UQW island can be nulled using the RG, which switches on and off the electrical connection of the island to the RC that plays a role of electronic reservoir. When zero or a positive bias is applied to the RG, the electrons from the SD channel compensate the positive charges in the UQW island and the detector is reset to its initial state.

We can thus distinguish two temporal regimes of operation. For sufficiently short times the second term on the right-hand side of (3) is negligible with respect to the first; combining with Equation (2) we can obtain the following expression of the total number of photons incident on the sample per unit time:(4)$$\Phi =\frac{WL}{\sigma_{0}}\frac{n_{LQW}}{n_{UQW}}\frac{1}{I_{SD}}{\left.\frac{dI_{SD}}{dt}\right|}_{t=0}$$

To derive this expression, we used the fact that $I_{SD}\left(t\right)={\Delta I}_{SD}\left(t\right)+I_{SD}\left(t=0\right)$. The quantity $\sigma_{0}$ is homogeneous to a surface and has been defined from the equality $\eta Tg=\sigma_{0}n_{UQW}$. The area $\sigma_{0}$ can be interpreted as the effective absorption cross section for a single electron [28,29], and, in the present device geometry it is fixed by the optical characteristics of the intersubband transition. Indeed, considering each electron in the island as a dipole oscillator, we can apply the approach of Tretyakov [17] obtaining the absorption cross section at resonance, $\sigma_{0}=\left(\frac{3\lambda^{2}}{8\pi n^{2}}\right)sin^{2}\left(45{}^{\circ}\right)[4\gamma \Gamma_{rad}/(\gamma {+\Gamma }_{rad}{)}^{2}]$. Here, $\Gamma_{rad}=ne^{2}f\omega_{21}^{2}/(12\pi \varepsilon_{0}m^{*}c^{3})$ is the radiation loss rate of the oscillating dipole [30] which is typically $\Gamma_{rad}\ll \gamma .$ Within this approximation, we obtain the expression $\sigma_{0}=\frac{e^{2}fsin^{2}\left(45{}^{\circ}\right)}{2n\gamma \varepsilon_{0}m^{*}c};$ then ${\eta g=\sigma }_{0}n_{UQW}/cos(45{}^{\circ})$ reproduces the expression of the intersubband absorption coefficient provided by Helm [27], within the factor of 2 which is due to the presence of the IG (see also the discussion on page 23 of Ref. [27]). Here, the transmission of the wedge is taken into account by the factor *T*.

Numerically, we have $\sigma_{0}=2.7\times 10^{-6}\;{µm}^{2}$ for the present sample. The ratio $n_{LQW}/n_{UQW}=1.25$ is also fixed in our case and in the order of unity; it has been determined from the electrical characteristics of the device (Figure 1c,d). Since all parameters of Equation (4) are known, *Φ* can be directly estimated by the electrical measurements on the device, such as the one shown in Figure 2c.

To avoid confusion, let us recall that, in our definition, *Φ* is the total number of photons incident on the sample; typically, sources are characterized by an intensity *I*, that is an optical power per unit surface. In our case, $\Phi =AI/ћ\omega_{12}$; where A is the absorbing area of the device, that is typically the area of the FG in Figure 2a. If one wants to reduce the photon number, it is natural to reduce *A*. However, in that case the relative change ${\Delta I}_{SD}/I_{SD}$ owing to photon absorption could be very small. In a typical device, *A* is of the same order of magnitude as the area of the SD channel. Within that assumption, from Equation (4), we can say that the ability of the detector to resolve very small photon numbers is provided by the area of the SD channel (factor *LW* in Equation (4)), as compared to $\sigma_{0}$, as well as by the ability of our measurement apparatus to detect small time variations of the SD current. Single photon-counting has been reported with small area CSIP devices [15]. Furthermore, the area $\sigma_{0}$ can be increased by photonic engineering of the IG [16]. Indeed, as shown in Ref. [19] for a general photonic architecture, $\sigma_{0}$ should be replaced by the absorbing cross section Σ of the structure, which can be substantially increased with respect to $\sigma_{0}$ by engineering of the radiation loss through antenna effects. Nano-antenna concepts can thus be very beneficial for keeping the photo-counting function of a CSIP, while increasing its overall quantum efficiency.

In previous studies [10,14], the figures of merit of the CSIP detectors have been estimated for the transient regime reported above. However, a valid comparison with other unipolar detectors is more direct in the saturation regime, where an equilibrium is reached between photo-excitation and the recombination of carriers. This is the typical regime of operation of QWIPs and QCDs, where the recombination time can be very fast, in the order of ps [31]. In the saturation regime, the time derivative on the left-hand side of Equation (3) is zero. The extra SD current is thus proportional to the photon number per unit of time; since the optical power incident on the device is $P=\Phi \hbar \omega_{12}$, we can define the detector responsivity in that regime, $R=\Delta I_{SD}/P$, to be:(5)$$R=\frac{e}{\hbar \omega_{12}}\eta Tg\frac{\tau }{\tau_{trans}}$$

Here, we introduced the SD transfer time ${\tau_{trans}=L^{2}/(µ}_{LQW}V_{SD})=57\;ns$ for $V_{SD}=50\;mV$. Clearly, the detector gain in this regime is $G=\tau /\tau_{trans}$; if τ can be made larger than $\tau_{trans}$, then high gains can be achieved and the responsivity of the detector can still be very high despite the low quantum efficiency from Equation (1).

It is interesting to note that this type of detector can be fully characterized by monitoring the SD current as a function of time; the time derivative of the SD current provides the incident photon number through Equation (4), and then the responsivity (5) can be determined in the saturation regime by measuring $\Delta I_{SD}$ for sufficiently long times. In that sense, the detector is auto-calibrated from knowledge of its electrical characteristics.

In the following, we characterize the detectors with a thermal source. For these measurements, we used a cryostat with a homemade cold finger where a Globar source was placed near the 45° facet of the device (Figure 3a). We further used a cryoshield cooled at 4 K to decouple the device from the ambient blackbody radiation during measurements. The impact of the source is already visible in the electrical characteristics of the device in Figure 2c. When the RG is on, an extra SD current appears for $V_{IG}<-0.6V$, for the IG voltage range where the UQW island has been fully isolated. The extra SD current is due to photoexcited holes that accumulate in the UQW island. An application of positive or zero RG voltage opens the island and the holes recombine through the RG contact; therefore, the optical signal is canceled.

Following Ref. [11], we applied a periodic modulation of the RG gate between 0 V and −2 V to follow the dynamics of the optical signal. The typical results are shown in Figure 3b where we recorded the $I_{SD}$ as a function of time with an oscilloscope (Tektronix TDS2024) for a $V_{IG}=-0.7\;V$ and $V_{SD}=50mV$. The reset bias was applied at time t=0, and lasted for about 2 ms.

As shown in the figure, in these measurements we varied the electrical power dissipated in the Globar, which resulted in a monotonic increase in the optical power incident on the device. At early times, we observed the expected linear increase in $I_{SD}$ as a function of time, while for very long times (*t*~2 ms) the current saturated at a constant value. (Note that the electrical measurements in Figure 2c were performed with a waiting time of 10 ms for each point, so that the SD current recorded was the saturated value). Equation (4) is used to determine the photon fluxes *Φ* for each measurement; the values obtained range from 2.5 × 10^10^ photons/s to 1.3 × 10^13^ photons/s (incident optical power from 0.65 nW to 322 nW for a photon energy of 133 meV). Similar measurements were performed with a smaller device with channel dimensions $W\times L=15\times 90\;µm^{2}$ (size of the IG was $15\times 25\;µm^{2}$), with a SD bias $V_{SD}=200\;mV$. In this case, we were able to detect smaller photon fluxes, from 4.7 × 10^9^ photons/s to 1.8 × 10^12^ photons/s (optical powers from 120 pW to 46 nW). Next, we evaluated the responsivity in the saturated regime from Equation (5). The results are plotted in Figure 3d as a function of the photon flux *Φ*. We observed a strong dependence of the responsivity as a function of *Φ*, with the responsivity increasing strongly for low values of the photon flux. The maximum responsivity was of the order *R*~720 A/W. The dependence *R*(*Φ*) is explained through the strong dependence of the hole lifetime with the number of photo-excited holes as discussed in Ref. [11]; namely, τ strongly decreases for high numbers of holes (high values of *Φ*). In Figure 3e, we plot the hole lifetime τ for each device as well as the values of the SD transit time $\tau_{trans}$. The corresponding values of the photodetector gain $G=\tau /\tau_{trans}$ were equal to 2.7 × 10^4^ for the 50 × 50 µm^2^ device and 1.3 × 10^4^ for the $15\times 25\;µm^{2}$ device, as indicated in the introductory paragraph. These values for the gain are several orders of magnitude larger than those typically observed in QWIPs or QCDs [3].

## 4. Conclusions and Perspectives

In this work, we report results on a CSIP detector based on GaAs/Al_0.33_Ga_0.67_As heterostructure designed for λ=9.3 µm wavelength operation. In the present case, we used a constant Aluminum content across the heterostructure instead of a graded one, making our design conceptually closer to that of QCD and QWIP detectors. While previously reported structures use metal gratings, here, we employ optical coupling through a 45° wedge facet, which has allowed us to provide a simplified and reliable expression for the incident photon flux *Φ* in the device (Equation (4)). With the help of this expression, we underline the “auto-calibrated” nature of such detectors; in the sense that all electrical measurements of the source-drain as a function of time $I_{SD}(t)$ such as the ones reported in Figure 3b, together with the knowledge of the detector geometry, are sufficient to determine the photon flux, and hence the optical power incident on the detector. In our expression, the measured photon flux is proportional to the ratio $LW/\sigma_{0}$, where *LW* is the area of the source-drain channel, and $\sigma_{0}$ the absorption cross section for a single electron that was previously introduced in nano-photonics [29] and explicated specifically for the case of intersubband transitions in Ref. [28]. For the latter case, it has been shown that $\sigma_{0}$ can be strongly enhanced using metamaterials [28,31,32]. Similar approaches also seem very appealing for CSIPs, which require small SD channel areas to operate as photon counters [14,15]. Increasing the absorption cross section $\sigma_{0}$ while reducing the channel area *LW* would allow the optimization of the CSIP photonic architecture for single photon-counting applications.

In the regime of saturation, where the optically induced extra SD current is no longer dependent on time, the detector operation is very similar to that of other quantum detectors, such as QWIPs and QCDs. In this regime, the CSIP devices can be viewed as a single period QCD with an integrated amplifier that can boost the photoconductive gain to an unprecedented level for such types of detectors. Another interesting feature of the CSIPs is that the responsivity is strongly dependent on the incident photon flux; such non-linearities can be further exploited for functions such as mixing [33]. Furthermore, the compact device architecture of the CSIP can become an asset in all integrated schemes such as the ones reported in Ref. [34], where both the source and detector originate from the same epitaxial wafer, especially in the case where the area of the SD channel is reduced in order to achieve larger sensitivity. All such integrated schemes are also very promising for molecular spectroscopy in the mid-infrared spectral range [35].

## Figures and Tables

**Figure 1 sensors-23-03635-f001:**
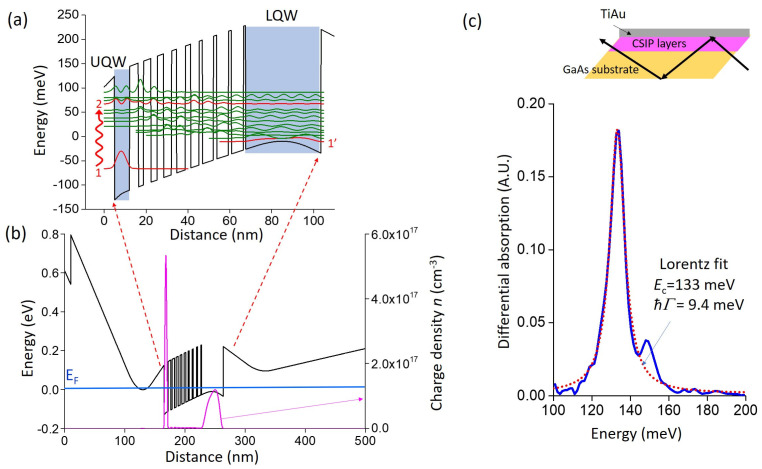
(**a**) Main sequence of GaAs/Al_0.33_Ga_0.67_As quantum well/barriers for the CSIP structure. The optical transition takes place between subbands 1 and 2 in the 7.2 nm wide upper quantum well (UQW). The 36 nm lower quantum well (LQW) is used as the source-drain channel. (**b**) Full band-structure simulation which includes the effect of the leftmost metal-semiconductor Schottky barrier, as well as the 1 × 10^18^ cm^−3^ modulation doped layers on each side of the structure. (**c**) Experimental differential absorption spectrum obtained in a multipass configuration (inset), as well as Lorentzian fit of the main absorption peak (dotted curve).

**Figure 2 sensors-23-03635-f002:**
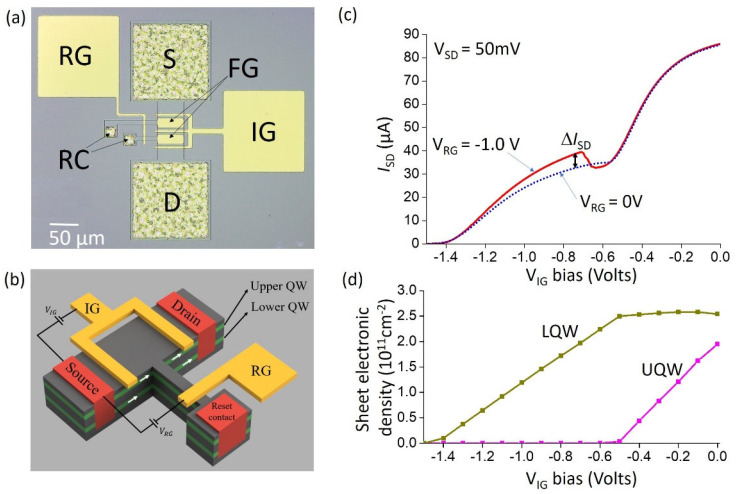
(**a**) Optical microscope photograph of a typical processed device, indicating the source (S) and drain (D) contacts, as well as various gates: IG: insulation gate, RG: reset gate, FG: floating gate. RC are the reset contacts. (**b**) 3D schematic diagram of the device showing the upper and lower quantum wells. Typically, the drain is connected to the ground. The FG is not shown for simplicity. (**c**) Source-drain current as a function of the IG bias, with $V_{SD}=50\;mV$ for two different biases on the RG. (**d**) Modelled electronic sheet density in the UQW and LQW as a function of the IG bias.

**Figure 3 sensors-23-03635-f003:**
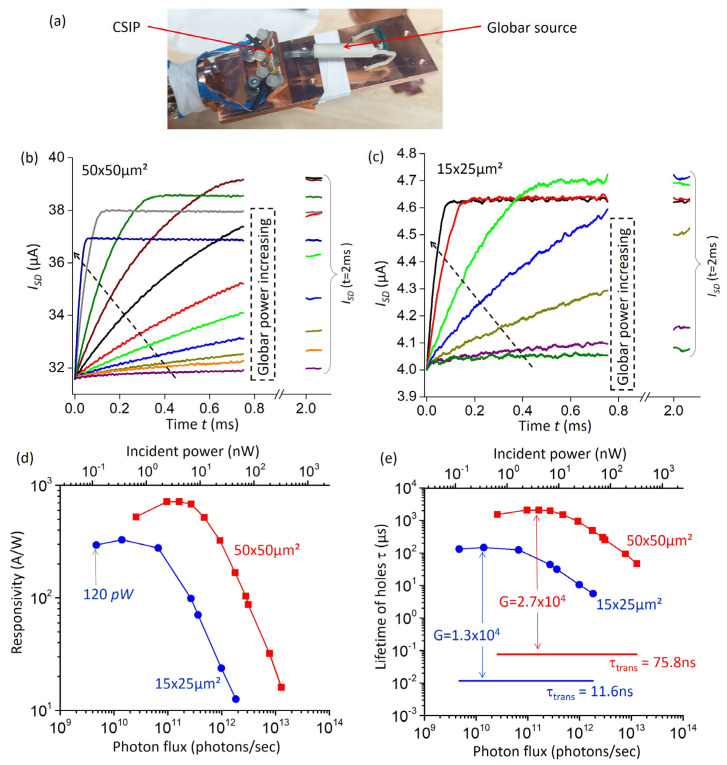
(**a**): Cold finger designed to study CSIP device with an external source at low temperature. The picture was taken with a Globar source placed next to the 45° facet of the CSIP device. (**b**) Source drain current I_SD_ as a function of time for a device identical to the one discussed in previous figures, $V_{IG}=-0.7\;V$ and $V_{SD}=50\;mV$. A reset gate bias $V_{RG}=-2\;V$ is applied at t=0 for approximately 2 ms after which the $V_{RG}$ is set to zero. For each measurement, we applied a different electrical power to the Globar source. The asymptotic values at the end of the reset period (*I*_SD_ (*t* = 2 ms)) are also indicated. (**c**) Source drain current *I*_SD_ as a function of time for a device with 15 µm × 25 µm absorbing region, and $V_{SD}=200\;mV$. (**d**) Responsivity at the saturation regime (Equation (5)) as a function of the photon flux that was determined from Equation (4) from the data in Figure 3b. We report the results for two different devices, where $X\times Y\;µm^{2}$ indicates the size of the FG. (**e**) Estimated lifetime of photoexcited holes in the saturation regime as a function of *Φ*. The horizontal lines indicate the SD transit time for each set of measurements/devices. We also indicate the maximum photoconductive gain measured for each device.

## Data Availability

The data presented in this study are available in the article.

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
