# Peer review of "Auto-Calibrated Charge-Sensitive Infrared Phototransistor at 9.3 µm"

_sensors, 2023, doi:10.3390/s23073635_

Round 1

Reviewer 1 Report

Overall, I have difficulties evaluating the novelty of the research work since most of the references are over 10 years. Many of the approaches in the manuscripts were referred to reference [11] which was published in 2009. The manuscript was not arranged using conventional methods, and hence, not many details on the experimental setup were given. I would suggest the authors thoroughly revise their manuscript before it can be evaluated for potential publication in Sensors.  

Author Response

R.1.1. Overall, I have difficulties evaluating the novelty of the research work since most of the references are over 10 years. Many of the approaches in the manuscripts were referred to reference [11] which was published in 2009.

A.1.1. In order to comply with the referee’s remark we have expanded the introduction with the full state of the art of CSIP detectors up to the present day, see lines 76-88 of the revised manuscript. Consequently, we have added new references to our list: [16-26] in the revised manuscript.

R.1.2. The manuscript was not arranged using conventional methods, and hence, not many details on the experimental setup were given.

A.1.2. We have provided more details on the experimental setup in the revised version: lines 130-149 (description of the differential absorption experiments), lines 154-175 (choice of light-coupling geometry) lines 180-183 (cryogenic conditions).   

Reviewer 2 Report

1.) The abstract conveys very less information about the study and gives an overview of the objective. But the results and how they achieved the results and methodology should also be included in the abstract.

2.) The introduction section is also vague. Proper and related literature should be reviewed. More information and background of the study should be provided.

3.) A full form should be mentioned whenever an abbreviation is used for the first time in the manuscript.

4.) Authors can also discuss some of the following latest advances in optical sensors:

Advanced Intelligent Systems, 4(1), 2100067. DOI: 10.1002/aisy.202100067

Photonics (Vol. 10, No. 1, p. 56). Multidisciplinary Digital Publishing Institute. DOI: 10.3390/photonics10010056

Ohodnicki, P. R., et al. "Fusion of Distributed Fiber Optic Sensing, Acoustic NDE, and Artificial Intelligence for Infrastructure Monitoring." Optical Fiber Sensors. Optica Publishing Group, 2022. DOI: 10.1364/OFS.2022.Tu1.1

IEEE Sensors Journal, 22(24), 23937-23944. DOI: 10.1109/JSEN.2022.3218797

5.) The purpose of the study is not clearly mentioned.

6.) What do the authors want to achieve/sense through the proposed mechanism should be discussed in detail.

7.) The write-up is not properly written and seems like a lab report rather than a research paper.

8.) Proper comparison should also be followed with similar mechanisms. 

Author Response

R2. 1. The abstract conveys very less information about the study and gives an overview of the objective. But the results and how they achieved the results and methodology should also be included in the abstract. 

A2.1. We have modified and extended the abstract in order to address the referee’s remark.

R2.2. The introduction section is also vague. Proper and related literature should be reviewed. More information and background of the study should be provided.

A2.2. We have expanded our introduction with the full state of the art of CSIP detectors up to the present day, see lines 76-88 of the revised manuscript. Consequently, we have added new references to our list: [16-26] in the revised manuscript.

R2.3. A full form should be mentioned whenever an abbreviation is used for the first time in the manuscript.

A2.3. This issue has been corrected: lines 36 and 40.

R2.4. Authors can also discuss some of the following latest advances in optical sensors:

  1. Advanced Intelligent Systems, 4(1), 2100067. DOI: 10.1002/aisy.202100067

b.Photonics (Vol. 10, No. 1, p. 56). Multidisciplinary Digital Publishing Institute. DOI: 10.3390/photonics10010056

  1. Ohodnicki, P. R., et al. "Fusion of Distributed Fiber Optic Sensing, Acoustic NDE, and Artificial Intelligence for Infrastructure Monitoring." Optical Fiber Sensors. Optica Publishing Group, 2022. D. DOI: 10.1364/OFS.2022.Tu1.1

IEEE Sensors Journal, 22(24), 23937-23944. DOI: 10.1109/JSEN.2022.3218797

A2.4. Our work is about quantum infrared detectors in the 9µm band (frequency 33 THz). We have stated the corresponding state of the art: references [1-5], [7-11], [14-26] of the revised manuscript. The references proposed above concern detectors/ sensors in completely different frequencies: telecom(a), 0.1-0.3 THz (b), telecom (c), 0.1-0.3 THz. We can not include them and comment them as they are completely outside the scope of our study.

R2.5. The purpose of the study is not clearly mentioned

A2.5. We have included a paragraph, lines 89-106, where we describe in details our objectives an position ourselves with respect to the state of art.

R2.6. What do the authors want to achieve/sense through the proposed mechanism should be discussed in detail.

A2.6) Lines 89-90 has been changed with respect to the original version: “Here we report a CSIP device designed to detect an infrared radiation with a wavelength of 9.3 µm.”

A2.7. The write-up is not properly written and seems like a lab report rather than a research paper.

R2.7. Where appropriate we have extended the discussion of our results in order to address the issue.

Our positioning with respect with previous studies is addressed in lines 89-106.

In lines   255-267 we provide more details about the estimation of the electron absorption cross-section in our sample, and in lines 299-305 we discuss the potential of nano-antenna concept in light of Eq.(4).  In lines 306-311 we discuss our method of comparison between CSIPs and other detectors.

R2.8. Proper comparison should also be followed with similar mechanisms.

A2.8. The comparison between CSIPs, QWIPs and QCDs has been extensively discussed in lines 306-365.

Round 2

Reviewer 1 Report

The authors have made major amendments to their manuscript and it is now carrying higher technical merit and thus can be accepted for publication. 

Reviewer 2 Report

The paper can be accepted in its current form.